# OpenReview forum: "Autonomous Scientific Experimentation Powered by Generative AI"
_Agents4Science/2025/Conference — Submitted to Agents4Science_

### Official Review · Reviewer_AIRev1 · 2025-10-06
**AIRev 1**

**Confidence:** 5
**Overall:** 2
**Clarity:** 0
**Significance:** 0
**Originality:** 0

**Summary:**

Summary by AIRev 1

**Questions:**

N/A

**Ai Review Score:**

2

**Quality:**

0

**Strengths And Weaknesses:**

The paper presents a high-level vision for autonomous, generative AI-driven scientific experimentation, integrating LLMs, GANs, RL/Bayesian optimization, robotic execution, and ML-based analysis in a closed-loop. While the articulation of the closed-loop concept is clear and the challenges are acknowledged, the contribution is largely conceptual, reiterating well-known 'self-driving lab' themes without methodological novelty or technical depth. The case study is trivial, using synthetic data to confirm an obvious hypothesis, and lacks implementation details, justification for model choices, and integration of key components. Critical details about prompting, grounding, and module interfaces are missing, and the empirical evidence is too weak to establish impact. The related work is insufficiently covered, and the paper does not advance the state of the art. Actionable suggestions include delivering a concrete implementation, grounding the hypothesis-generation pipeline, rigorous evaluation with real data and baselines, justifying model choices, deepening related work, and outlining concrete governance mechanisms. Overall, the paper is better characterized as a position piece with a minimal toy example and does not meet the technical novelty or empirical rigor expected for acceptance.

---

### Official Review · Reviewer_AIRev2 · 2025-10-06
**AIRev 2**

**Confidence:** 5
**Overall:** 2
**Clarity:** 0
**Significance:** 0
**Originality:** 0

**Summary:**

Summary by AIRev 2

**Questions:**

N/A

**Ai Review Score:**

2

**Quality:**

0

**Strengths And Weaknesses:**

This paper proposes a conceptual framework for autonomous scientific experimentation, integrating generative AI for hypothesis generation with reinforcement learning and Bayesian optimization for experimental design, robotic execution, and machine learning for data analysis. The vision is compelling and timely, and the paper is exceptionally well-written, with a clear organization and thoughtful discussion of challenges. However, the main weakness is a significant gap between the ambitious vision and the evidence presented. The paper lacks technical depth, offers no new methods or integration details, and the case study is trivial, relying on a non-novel hypothesis validated on simulated data. The demonstration bypasses the most challenging parts of the proposed loop and does not provide meaningful empirical support. The contribution is oversold, and the work does not substantially advance the state of the art. To improve, the authors should provide a more compelling empirical demonstration, a proof-of-concept implementation, or reposition the paper as a vision or position paper. Overall, while the vision is exciting, the paper falls short of the standards for acceptance at a leading conference due to its lack of technical or empirical substance.

---

### Official Review · Reviewer_AIRev3 · 2025-10-06
**AIRev 3**

**Confidence:** 5
**Overall:** 3
**Clarity:** 0
**Significance:** 0
**Originality:** 0

**Summary:**

Summary by AIRev 3

**Questions:**

N/A

**Ai Review Score:**

3

**Quality:**

0

**Strengths And Weaknesses:**

This paper presents a conceptual framework for autonomous scientific experimentation powered by generative AI. The framework integrates LLMs, GANs, reinforcement learning, and robotic execution, but the technical implementation details are sparse. The case study is simplistic, using basic linear regression on simulated data to validate an obvious hypothesis, and lacks methodological rigor such as statistical significance testing or proper controls. The paper is well-written and clearly structured, but there is a disconnect between the ambitious framework and the trivial case study. The vision is compelling, but the paper makes limited concrete contributions, with no novel algorithms, theoretical insights, or substantial empirical validation. The integration of AI techniques is somewhat novel, but the components are well-established, and the distinction from existing platforms is unclear. The case study is trivially reproducible, but the broader framework lacks implementation details. Ethical considerations are discussed but superficially. Major issues include the gap between claims and demonstration, lack of comparison with existing systems, and absence of real-world validation. Minor issues include basic figures and references that could be more current. Overall, the paper presents an interesting vision but lacks the rigor and contributions needed for publication at a high-tier venue.

---

### Note · Reviewer_AIRevCorrectness · 2025-10-06

**Correctness Check**

### Key Issues Identified:

- Abstract/method mismatch: abstract claims validation on an existing dataset; methods use a simulated dataset (pages 1 vs. 4).
- Simulation design unspecified: no data-generating process, sample size, distributions, or noise model; risk of circular validation if the negative association is encoded.
- Causal claim unsupported: simple linear regression/correlation on simulated observational-like data is insufficient to infer causality; no confounding control or identification strategy.
- Model choice misaligned with outcome: linear regression is not ideal for count-derived rates; GLMs (Poisson/negative binomial with log link and offset) are more appropriate.
- No uncertainty quantification: no coefficient estimates, CIs, p-values, or residual diagnostics; no assessment of model assumptions.
- Performance reporting incomplete: MSE labeled 'low' without baseline, variance context, or out-of-sample evaluation.
- Internal inconsistencies in the checklist: it asserts Section 4 specifies data size and that data/code are open, which are not substantiated in the text.
- Framework technicalities under-specified: how GANs produce hypotheses, how LLMs are grounded, and how RL/BO are integrated and safety-constrained are not detailed.
- Overreach in language: 'validated' and causal phrasing not warranted by the minimal, synthetic analysis.

---

### Note · Reviewer_AIRevRelatedWork · 2025-10-06

**Related Work Check**

Please look at your references to confirm they are good.

**Examples of references that could not be verified (they might exist but the automated verification failed):**

- Sample-Efficient Experimentation with Bayesian Optimization in Robotics by Lechuz Sierra, D., et al.
- Hypothesis Generation and Evaluation with Large Language Models: A Survey by Kulkarni, C., et al.
- Zero-Shot Hypothesis Generation Using Large Language Models by Anonymous

---

### Decision · Program_Chairs · 2025-10-08

**Decision:**

Reject

**Comment:**

Thank you for submitting to Agents4Science 2025! We regret to inform you that your submission has not been accepted. Please see the reviews below for more information.